# Progress in Diagnosing Primary Ciliary Dyskinesia: The North American Perspective

**DOI:** 10.3390/diagnostics11071278

**Published:** 2021-07-16

**Authors:** Michael Glenn O’Connor, Amjad Horani, Adam J. Shapiro

**Affiliations:** 1Pediatric Pulmonary Medicine, Vanderbilt University Medical Center, Nashville, TN 37232, USA; 2Department of Pediatrics, Washington University School of Medicine, St. Louis, MO 63130, USA; horani_a@wustl.edu; 3Department of Cell Biology and Physiology, Washington University School of Medicine, St. Louis, MO 63110, USA; 4Pediatric Pulmonary Medicine, McGill University Health Centre Research Institute, Montreal, QC H4A 3J1, Canada; adam.shapiro.med@ssss.gouv.qc.ca

**Keywords:** primary ciliary dyskinesia, PCD, diagnostic guidelines, North America

## Abstract

Primary Ciliary Dyskinesia (PCD) is a rare, under-recognized disease that affects respiratory ciliary function, resulting in chronic oto-sino-pulmonary disease. The PCD clinical phenotype overlaps with other common respiratory conditions and no single diagnostic test detects all forms of PCD. In 2018, PCD experts collaborated with the American Thoracic Society (ATS) to create a clinical diagnostic guideline for patients across North America, specifically considering the local resources and limitations for PCD diagnosis in the United States and Canada. Nasal nitric oxide (nNO) testing is recommended for first-line testing in patients ≥5 years old with a compatible clinical phenotype; however, all low nNO values require confirmation with genetic testing or ciliary electron micrograph (EM) analysis. Furthermore, these guidelines recognize that not all North American patients have access to nNO testing and isolated genetic testing is appropriate in cases with strong clinical PCD phenotypes. For unresolved diagnostic cases, referral to a PCD Foundation accredited center is recommended. The purpose of this narrative review is to provide insight on the North American PCD diagnostic process, to enhance the understanding of and adherence to current guidelines, and to promote collaboration with diagnostic pathways used outside of North America.

## 1. Summary

Primary ciliary dyskinesia (PCD) is a rare disease with an estimated prevalence of 1 in 15,000 individuals [1,2]. Despite improved understanding of the clinical phenotype, PCD remains under-recognized. The PCD Foundation (PCDF) estimates that only about 10% of people with PCD have been definitively diagnosed and followed at a PCD clinical center. This discrepancy is likely due to low disease recognition by clinical providers and limited access to specialized diagnostic services in many clinical centers across the US and Canada.

In 2018, a panel of experts, sponsored by the American Thoracic Society (ATS), published a PCD diagnostic pathway focused on techniques most readily available in the US and Canada [3]. While these guidelines recognize the utility of highly specialized centers of excellence, accredited in the PCDF Clinical and Research Center Network (CRCN), they also consider a comprehensive approach for clinicians who are not a part of the PCDF CRCN.

In 2017, the European Respiratory Society (ERS) also published guidelines based on pre-defined questions relevant for the diagnosis and clinical care of PCD patients [4]. Differences in the ATS and ERS guidelines have previously been highlighted [5] and reflect the differences in diagnostic testing available in North America and Europe.

The ATS Clinical Practice Guideline for diagnosing PCD recommends that, when possible, appropriate patients with a compatible clinical PCD phenotype first undergo nasal nitric oxide (nNO) measurement, as an easy and well-established tool for PCD screening. Individuals with low nNO values can then be further evaluated using confirmatory PCD tests, including extended panel genetic testing or ciliary ultrastructural evaluation by transmission electron microscopy (TEM). The guideline also recognizes that nNO testing may not be readily available or feasible when patients are young (and thus uncooperative). In these cases, extended panel genetic testing is recommended as the first-line test for PCD diagnosis. Notably, some PCD testing modalities used outside of North America, including ciliary beat pattern analysis by high speed videomicroscopy, [6] was not adopted by the ATS guidelines.

In addition to guideline recommendations, we briefly present the evolving network of PCD specialty centers being established across the US and Canada through the PCDF CRCN. Through a standardized accreditation process, participating centers adhere to common benchmarks for PCD diagnosis and therapies [7], while also participating in a PCD Foundation clinical patient registry, which will eventually aid with PCD clinical trials across North American centers.

## 2. Recognition of the Clinical Phenotype

PCD is a disease of impaired motor cilia function associated with a growing list of causative genes. Most pediatric PCD patients exhibit several key clinical features including: (a) persistent, year-round wet cough that starts in the first year of life, (b) persistent, year-round nasal congestion that also starts in the first year of life, (c) the presence of organ laterality abnormalities, and (d) unexplained neonatal respiratory distress in infants born at term gestation [8]. While PCD patients often display otitis media with persistent effusion, chronic bronchitis, and recurrent pneumonia, these issues are prevalent in many other children presenting for respiratory care and thus do not readily assist physicians in discerning which patients should undergo further PCD investigations. However, the complete absence of middle ear disease or recurrent lower respiratory tract infections makes PCD less likely. The ATS PCD diagnostic guidelines recommend that at least two of the key clinical PCD features be present for the pre-test probability to be high enough to proceed with PCD diagnostic testing [3,8].

Unlike other overlapping pediatric respiratory conditions, PCD usually presents with persistent symptoms on a daily, non-seasonal basis; the symptoms never completely resolve, even with prolonged antibiotic courses. The persistent wet cough and rhinorrhea often appear before 6 months of age, though many families report onset at birth. Neonatal respiratory distress, despite term birth, is seen in at least 80% of PCD cases and can have a delayed onset (median onset at 12 h of life, with a range of 0 to 72 h of life), accompanied by lobar atelectasis on chest radiography [9]. The duration of neonatal respiratory distress in PCD is often prolonged, with a median of 14 days [8,9], and some cases require supplemental oxygen therapy for several months.

In adult PCD populations, persistent wet cough and nasal congestion on a daily basis remain key clinical features, though the age of symptom onset and presence of neonatal respiratory distress are often forgotten in these cases [10]. Chronic sinusitis with polyposis and male infertility, as well as female subfertility, are also frequently reported by adults with PCD; additionally, bronchiectasis is universally present in the adult PCD populations.

Organ laterality defects are also common in PCD, due to the motility defects of the embryonic monocilia, which appear during early embryogenesis. However, organ laterality defects only occur in 50% of individuals with PCD and can result in a variety of organ arrangements, including situs inversus totalis (mirror image organ arrangement), situs ambiguus (a mix or left-right organ arrangement, often with congenital heart defects), isolated organ laterality defects (such as dextrocardia), polysplenia, interrupted inferior vena cava, and others [11] (Figure 1). When situs ambiguus occurs with complex congenital heart lesions, the label “heterotaxy” is often used, though situs ambiguus organ arrangements can similarly occur with mild congenital heart defects (such as ventricular septal defects). The terms situs ambiguus and heterotaxy are often employed interchangeably, which may result in confusion amongst clinicians.

## 3. PCD Diagnostic Testing

The historical diagnostic standard for PCD has been ciliary ultrastructural analysis by transmission electron microscopy (TEM) and while classic ultrastructural defects can yield a positive diagnosis, TEM is normal in approximately 30% of PCD cases. Genetic testing for PCD continues to improve with an expanding list of causative genes, but this testing is also inconclusive in 20–30% of PCD cases. Nasal nitric oxide (nNO) testing in an appropriate patient can further validate suspicion for PCD, but low nNO values should be validated with either genetic or TEM testing. Only a classic TEM ultrastructural defect or a positive genetic test can definitively confirm a diagnosis of PCD, in isolation from other test results; however, in the absence of confirmatory TEM or genetic testing, repeated low nNO values in an appropriate clinical phenotype can yield a provisional diagnosis of PCD.

### 3.1. Nasal NO Testing

Nitric oxide (NO) is a colorless, odorless gas produced in the upper and lower respiratory epithelium. It has diverse effects in the respiratory system, including vasodilation, bacterial killing, and the modulation of inflammation [12,13]. Fractional exhaled NO from the lower airways (FeNO) is normally upregulated in times of infection and inflammation (as often seen in poorly controlled asthma), while nasal NO is mainly produced in the paranasal sinuses.

Nasal nitric oxide (nNO) measurement represents a simple, non-invasive way to screen patients for PCD, with results immediately available at the time of testing. For unclear reasons, nNO values are distinctly reduced in individuals with PCD [14]. Other conditions that may have overlapping low nNO values include cystic fibrosis, diffuse pan-bronchiolitis, acute viral respiratory infection, and certain forms of primary immunodeficiency [15,16,17,18,19,20]. Thus, these conditions should always be considered and investigated if relying on low nNO as the sole diagnostic evidence for likely PCD (i.e., negative genetic testing and normal TEM analysis). When nNO values are repeated over time and remain persistently low, they can yield a provisional diagnosis of PCD, but cannot fully confirm the diagnosis and must be taken in context with other PCD diagnostic test results. It should be noted that only chemiluminescence NO devices have been prospectively validated for use in PCD investigations, with results from significantly less expensive electrochemical NO devices lacking robust studies in large PCD populations [21,22].

Cross-sectional and longitudinal observations have shown that ≥90% of individuals with PCD have nasal NO levels below 77 nL/min [18], when measured per a standardized protocol. Pooled sensitivity and specificity estimates for nNO < 77 nL/min are 96% and 96%, respectively, in cases of PCD that were confirmed using TEM and/or genetic analysis and when cystic fibrosis has been ruled out [23]. Low nasal NO values should be confirmed with at least one additional test on a separate visit, with patients at their baseline health, and without evidence of viral respiratory infection.

Around 5–10% of PCD cases have nNO values above the established cut-off of 77 nL/min, specifically those associated with mutations in radial spoke or central apparatus proteins [23,24]. These include cases due to genetic variants in *RSPH1*, which are associated with normal ciliary ultrastructure by TEM and nNO values of 100 to 300 nL/min [18] (Table 1).

Successful nNO testing requires a patient be cooperative enough to blow into a resistor device and achieve steady state NO measurement. Children under 5 years old often cannot perform this maneuver, so nNO testing may not be appropriate in preschool-aged children. Tidal breathing nNO measurements for patients 2–5 years old are feasible and are currently under investigation; however, diagnostic cut-off values are undefined in this age group. According to ATS guidelines, in patients with an appropriate PCD clinical phenotype and one nNO value <77 nL/min, a repeat measurement is recommended at least 2 weeks later [44]. With repeat low nNO values and negative cystic fibrosis testing, a presumptive diagnosis of PCD can be made and appropriate PCD therapies commenced. However, the availability of nNO testing is still hampered by the lack of clinical regulatory approval in North America; thus, most nNO tests are performed under research protocols in certified PCD centers, which are not reimbursed by insurance plans.

### 3.2. PCD Genetic Testing

PCD is a genetically heterogenous disease caused by variants of more than 50 genes. This number will probably increase as more gene mutations are associated with PCD. The most frequently reported mutations occur in *DNAI1* and *DNAH5*, which were the first implicated PCD genes [45,46]. The high number of pathogenic variants in different genes associated with PCD is a derivative of the complexity of the motile cilium, which is composed of hundreds of structural proteins and an unknown number of cytoplasmic factors that are required for ciliary assembly. Most PCD gene mutations are inherited in an autosomal recessive fashion. However, other inheritance modes occur, including x-linked inheritance in the assembly factor *PIH1D3* and autosomal dominant inheritance in mutations of the forked head transcription factor *FOXJ1* [33,47,48]. It is believed that mutations in known PCD-associated genes account for approximately 70–80% of PCD cases [49] (Figure 2).

Numerous commercial PCD genetic panels are available, varying in the number of genes covered and the employed sequencing technology. Most panels include the next generation sequencing analysis of at least 30 genes, with some offering more than 40 genes, plus deletion/duplication analysis. Panels covering greater numbers of PCD genes will have greater detection rates [50] (Figure 2). It is expected that the routine use of newer sequencing methods, such as whole exome sequencing (which offers better coverage), will likely replace panels with limited numbers of genes [51].

Although the increased availability of genetic tools is transformative for PCD diagnosis, interpreting the results of these tests may be challenging, due to the high number of variants of unknown significance (VUS), which often occur when there is discrepancy in the prediction tools used to determine pathogencity or when mutations were not previously reported to be pathogenic. PCD diagnosis requires two pathogenic variants be found in a single known PCD gene, on two opposite chromosomes (in trans) or on one allele, in the case of x-linked or autosomal dominant forms (*PIH1D3*, *RPGR*, *OFD1*, or *FOXJ1).* Pathogenicity of VUS results should be evaluated on a case-by-case basis, after consultation with either a geneticist or a PCD specialty center; VUS results cannot be assumed as disease-causing for diagnostic purposes.

### 3.3. Cilia Ultrastructure Evaluation

Traditionally, the use of TEM to analyze ciliary ultrastructure was considered the gold standard for a diagnosis of PCD. Motile ciliary cross sections show a typical 9 + 2 arrangement (Figure 2) that represents 9 outer microtubule doublets surrounding a central pair. Outer dynein arms (ODA) and inner dynein arms (IDA) extend from the microtubule, acting as the motors powering ciliary movement. [52]. Radial spokes and nexin links provide stability to beating ciliary axonemes.

Procuring and processing TEM samples requires significant expertise. From sample collection considerations (i.e., nasal versus endobronchial scraping as opposed to “pinch” tracheal biopsies) to sample processing considerations (i.e., choice of fixatives and skill of technicians in choosing multiple ciliary tufts across the sample) the TEM image quality can vary greatly. Interpretation is similarly challenging and requires evaluation by pathologists with extensive experience examining ciliary ultrastructure. To address these challenges, a more recent international consensus guideline focused on reporting TEM results in the diagnosis of PCD, which included minimal criteria for acceptable TEM studies and the creation of various diagnostic classes of TEM defects. Per this protocol, only “hallmark” defects (class 1: absent ODA, absent ODA+IDA, and absent IDA with microtubular disorganization) are reliably diagnostic of PCD [53]. Ultimately, 30% of genetically proven cases of PCD do not have a hallmark defect on TEM [1], due to the limited sensitivity of EM in identifying small structural changes in the electron dense ciliary axoneme. This is especially true when identifying changes affecting the central pair, radial spokes, or inner dynein arms [54].

A promising diagnostic test using immunofluorescent-labeled antibody microscopy has been adopted in some European PCD centers [55]. Immunofluorescent antibody microscopy has the advantage of being more readily available than TEM, costing less per sample, and is possibly more sensitive than TEM depending on the panel of antibodies used. However, the diagnostic accuracy of immunofluorescent antibody testing has not been firmly established in PCD [55]. Most local pathology laboratories also possess the technical knowledge to process and interpret immunofluorescence antibody samples, as this technique is currently used in a variety of human diseases. Immunofluorescent antibody microscopy is currently not available for clinical use in North American centers, though this technique is being considered at several centers.

### 3.4. Ciliary Motility and Highspeed Video Microscopy

Mutations associated with PCD often result in changes to ciliary beat frequency and waveform. The use of a high-speed video microscopy (HSVA) recording for ciliary waveform analysis is a validated PCD diagnostic tool, which is mostly available in large European centers [4]. Investigations have shown high sensitivity and specificity for HSVA, compared to ultrastructural defects on TEM, when performed by specialty centers with extensive experience in this testing modality [56]. HSVA results should be confirmed by repeat analyses on several occasions or on cultured ciliated airway cells, which requires expertise that may not be available in some centers. Further research is needed to compare the diagnostic accuracy of HSVA against PCD cases with rare genetic causes, as subtle waveform changes may not be easily detected by HSVA [4,57].

HSVA testing has not gained widespread adoption by PCD centers in North America, outside of limited research settings [58]. Similar to nNO testing, a normal cilia waveform analysis does not preclude a diagnosis of PCD, as some mutations may result in normal or near-normal movement [58]. The use of standard light microscopy for the calculation of ciliary beat frequency or waveform analysis, without high-speed video recording, is not an acceptable method to either diagnose or rule out PCD and should be avoided. Future standardization of HSVA practices, including training in image capture and the advancement of automated or semi-automated analysis algorithms, may allow increased use of high-speed video microscopy analysis as a diagnostic tool for PCD in North American centers.

## 4. Summary Considerations for PCD Diagnostic Testing in North America

The ATS PCD diagnostic guidelines (Figure 3) take into consideration the diversity and geography of North American medical centers, where PCD diagnostic testing often occurs. With the number of separate, academic medical centers across the US and Canada, PCD diagnostic testing cannot be performed at only a few, select, national centers of excellence, as is common in Europe. Furthermore, due to financial and insurance regulations, patients cannot readily cross state/provincial borders for PCD testing if services are lacking in their jurisdictions. Thus, the feasibility and availability of PCD diagnostic tests factored highly into the ATS recommendations.

A central focus of the ATS recommendations is careful recognition of the clinical phenotype, specifically focusing on the clinical features in the patient history that can help distinguish those individuals more likely to have PCD from other respiratory conditions. In starting with phenotype recognition, the ATS guidelines promote the active participation of clinicians in the diagnosis of PCD, recognizing that it is possible to make a diagnosis of PCD through the astute recognition of clinical symptoms and careful interpretation of genetic results. While genetic testing is widely available at most medical centers in the US and Canada, this testing is not always available outside of North America, mainly due to cost and access issues. However, genetic testing samples can be easily obtained through blood, saliva, or buccal samples; thus, the ATS guidelines present a pathway for improved PCD diagnosis in middle- and low-income countries, if improved access to genetic testing is provided.

The ATS guidelines also recognize the benefits of a network of accredited PCD diagnostic centers with expertise in specialized testing. While nNO testing still requires access to a chemiluminescence analyzer, these are becoming more widespread through the efforts of the PCD Foundation to expand the number of accredited centers in the PCDF CRCN. In addition, published technical guidelines for standardized nNO testing now exist [44], which adds uniformity to the test results that are completed at centers throughout the US and Canada. Challenges with ciliary TEM reliability and feasibility do exist in North American academic centers [59], despite this testing modality being well established in pathology laboratories and used for a multitude of other diseases. While efforts are ongoing to address challenges with the testing, the ATS guidelines recommend TEM analysis as a second-line test when genetics are inconclusive. This represents a departure from previous European Respiratory Society recommendations [5,60], though recent advances in cryo-electron tomography show promise for ultra-high resolution ciliary images, which may be incorporated into future clinical diagnostic testing [61].

In conclusion, while the focus of the ATS guidelines was the US and Canada, many of the diagnostic challenges described in this manuscript exist in other areas of the world. The guidelines recognize a pathway to PCD diagnosis with careful recognition of the phenotype and access to genetic testing, making a diagnosis of PCD a reality for many geographic locations without access to specialized testing, such as ciliary structural and functional analysis or nNO testing. However, the guidelines also promote a network of accredited centers with expertise in more specialized testing for those PCD clinical cases that are more challenging. In this recognition, the ATS guidelines provide a platform for collaboration with specialized PCD diagnostic centers throughout the world, including Europe. From these perspectives, the ATS guidelines have a broad reach to inform the diagnosis of PCD in many parts of the world.

## 5. PCD Foundation CRCN Overview

Over the last 15 years, the PCD Foundation has worked closely with clinicians and researchers to uncover much of the PCD medical knowledge possessed today. Building upon this success, the Foundation is establishing PCD centers of excellence across the US and Canada. Most centers are tied to large academic medical institutions where patients receive unified care from pulmonologists, otolaryngologists, cardiologists, neonatologists, pathologists, and geneticists. Each center also has well-established ancillary supports through nursing, social work, genetic counseling, and respiratory/physical therapy.

Through a detailed accreditation process, clinical centers are certified in nNO measurement with chemiluminescence devices, genetic testing with interpretation, and TEM analysis of ciliary ultrastructure. All PCD diagnostic testing and interpretation is performed using standardized protocols in close collaboration between the Foundation and expert PCD physicians leading each center. Over 40 pediatric and adult locations are accredited in the PCDF CRCN). (Figure 4). Each center closely collaborates on many aspects of PCD clinical care with an annual academic conference where research products and clinical innovations are presented.

One of the ongoing efforts of this network is the establishment of an online, clinical patient registry that allows tracking of PCD clinical data and serves as a central location from which future research endeavors can be launched. These efforts will inform future diagnostic algorithms by better describing phenotypic, genotypic, and diagnostic testing diversity in patients with PCD. With the goal to establish at least one accredited PCD clinical center in each state/province within the US and Canada, the Foundation will provide accurate, reliable, and affordable PCD diagnostic testing to all afflicted patients and an improved support network for medical providers outside of the network.

## 6. Conclusions

The 2018 ATS PCD diagnostic guidelines describe an approach to diagnosing patients with PCD, considering the feasibility and availability of diagnostic testing in the US and Canada. The growing PCDF CRCN exists to promote the accurate diagnosis of PCD both within and outside the network. Nasal nitric oxide testing is recommended as a first-line PCD screening tool, but initial genetic testing is certainly reasonable in clinical situations with a high likelihood of PCD or when nNO testing is not available. Although these guidelines represent the current state of PCD diagnostic testing in the US and Canada, international collaboration has the potential to expand and improve the diagnosis and treatment of this rare lung disease.

## Figures and Tables

**Figure 1 diagnostics-11-01278-f001:**
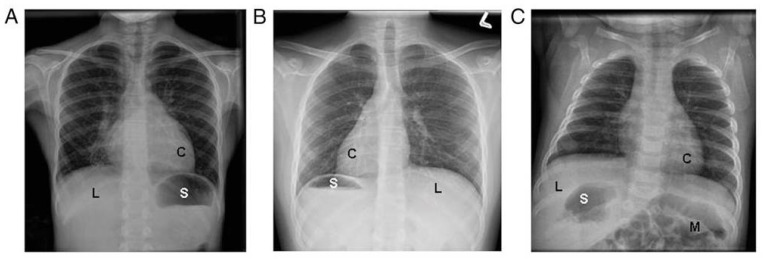
Examples of laterality defects on radiology imaging: (**A**) situs solitus, (**B**) situs inversus totalis, and (**C**) situs ambiguus. (C = cardiac apex; L = liver; M = intestinal malrotation; S = stomach). Reprinted with permission from ref. [11]. Copyright 2014 Shapiro AJ, Davis SD, Ferkol T, Dell SD, Rosenfeld M, Olivier KN, et al.

**Figure 2 diagnostics-11-01278-f002:**
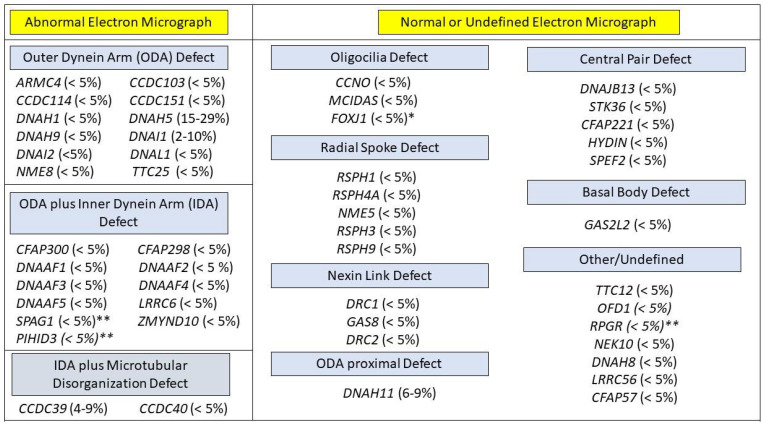
PCD genotypes, grouped by expected ciliary ultrastructure on electron micrograph (EM). Genotypes with an x-linked mode of inheritance are labeled with **. Genotypes in with an autosomal dominant mode of inheritance are labled with *. All other genotypes have an autosomal recessive mode of inheritance. Approximate prevalence of each genotype appears in paretheses.

**Figure 3 diagnostics-11-01278-f003:**
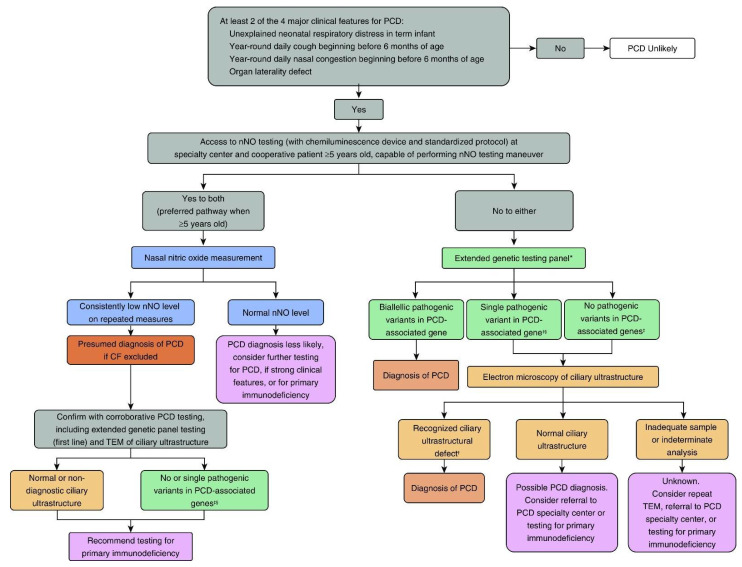
American Thoracic Society (ATS) PCD Diagnostic Guidelines. * Genetic panels testing for mutations in >12 disease associated PCD genes, including deletion/duplication analysis. † Known disease-associated TEM ultrastructural defects include outer dynein arm defects, outer dynein arm plus inner dynein arm defects, and IDA defects with microtubular disorganization. ‡ In genes associated with autosomal recessive trait. § Or presence of variants of unknown significance. Adapted from American Journal of Respiratory and Critical Care Medicine. CF = cystic fibrosis; nNO = nasal nitric oxide; PCD = primary ciliary dyskinesia; TEM = transmission electron microscopy. Reprinted with permission Ref. [24]. Copyright © 2021 American Thoracic Society.

**Figure 4 diagnostics-11-01278-f004:**
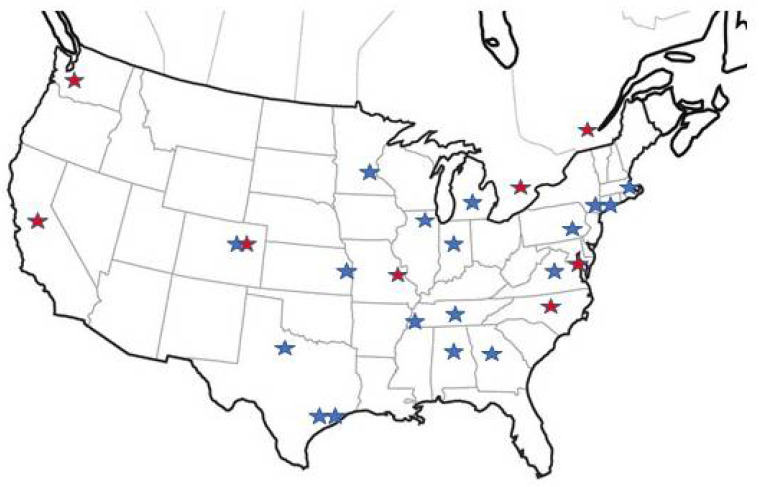
Map of PCD Foundation accredited clinical centers in the United States and Canada. Location of centers represented by stars. Red stars represent the original PCD centers of the Genetic Disorders of Mucociliary Clearance. Blue starts represent new centers that have been added since the formation of PCDF CRCN and creation of diagnostic guidelines.

**Table 1 diagnostics-11-01278-t001:** PCD genotypes grouped by reported nasal nitric oxide (nNO) values. Reported case numbers of specific genes with nNO > 77nl/min are presented in parentheses.

nNO Routinely ≤ 77 nL/min	Limited Cases with nNO > 77 nL/min	nNO Routinely > 77 nL/min
ARMC4	DNAI2	DNAH9 (9 cases) [25,26]	CCDC103 [27]
CCDC39	DNAJB13	TTC12 (2 cases) [28]	RSPH1 [29]
CCDC40	DNAL1	RPGR (<10 cases) [30,31]	
CCDC114	DRC1	CCNO (2 cases) [32]	
CCDC151	DRC2	FOXJ1 (4 cases) [33]	
CFAP57	HYDIN	NEK10 (1 case) [34,35]	
CFAP298	LRRC6	GAS2L2 (2 cases) [36]	
CFAP300	MCIDAS	GAS8 (1 cases) [37]	
DNAAF1	NME5	STK36 (1 case) [38]	
DNAAF2	NME8	CFAP221 (3 cases) [39]	
DNAAF3	OFD1	SPEF2 (1 cases) [40,41,42]	
DNAAF4	PIHID3	LRRC56 (1 case) [43]	
DNAAF5	RSPH3		
DNAH1	RSPH4A		
DNAH5	RSPH9		
DNAH8	SPAG1		
DNAH11	TTC25		
DNAI1	ZMYND10

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
