# Peer review of "Progress in Diagnosing Primary Ciliary Dyskinesia: The North American Perspective"

_diagnostics, 2021, doi:10.3390/diagnostics11071278_

Round 1

Reviewer 1 Report

This is a comprehensive review of PCD, focusing on the North American perspective.  It provides and overview of the clinical guidelines, and the investigations of a patient suspected or diagnosed with PCD. 

Although the focus of the review is on the North American experience, this manuscript appears to have an over-emphasis on that aspect.  Bringing in more elements from the European guidelines, approach and why they differ may also provide some balance.  Any experience outside of Europe and North America?  Furthermore, outlining regions in the North America, and under-served areas, and strategies to overcome may be also provide future next steps in a disorder where the research is biased towards a few ethnicities may address equity, diversity and inclusion.

Patient representation from the PCDF and or patients themselves, and outlining their experience of the North American approach might also be useful.

Reviewer 2 Report

The narrative review summarizes the current clinical practice recommendations proposed in 2018 by the American Thoracic Society (ATS) for the diagnosis of  Primary Ciliary Dyskinesia (PCD). Furthermore, it presents the evolving network of PCD specialty centers for PCD diagnosis and therapies being established across the US and Canada.

This review represents the starting point that lays the foundations for clinical trials, future studies of genetics and genotype-phenotype correlation and for possible revision of the current recommendations.

The review is well written and it analyzes the crucial role played by availability and feasibility of PCD diagnostic tests in the North America on the ATS recommendation.

In some points the manuscript is often very general and is missing the comments of authors on the strengths and limitations of the guideline recommendations based on current knowledge.

Abstract

In the abstract the purpose of this narrative review is missing. Add it and state that your study is a narrative review.  

Page 1 line 19 “positive nNO results”.  “Low levels of  nNO” is better.

Recognition of the Clinical Phenotype.

Page 2 line 81 “Neonatal respiratory distress despite term birth is seen in at least 80% of PCD cases and can have a 79 delayed onset (median onset at 12 hours of life, with a range of 0 to 72 hours of life) accompanied by lobar atelectasis on chest radiography” add reference.

Page 2 lines 87 and 97 remove brackets.

In the manuscript there is a frequent use of brackets, I suggest to delete some of them.

Introduction to PCD diagnostic testing

Page 3 line 101. Paragraph 3 called “Introduction to PCD diagnostic testing” is too small to be an independent paragraph. Call it “PCD diagnostic testing”. Remove the subdivision into paragraphs 4-5-6-7 and include their content in the only paragraph 3 entitled “PCD diagnostic testing”.

Finally, there is an important topic about diagnosis testing that is left out such as Immunofluorescence. A short section on it needs to be add.

Page 4 line 138 add reference

Page 4 line 148 “(and negative cystic fibrosis testing)” remove brackets

Page 4 line 149 specify according ATS recommendations  

Page 6 Figure 2 shows well-known electron microscopy images. It is superfluous in the article.

Page 5 Table 2 could be made clearer. There are genes marked with asterisks that are not reported in the legend. Use the same character for genes with different mode of inheritance and mark them with different asterisks, that you explain in the legend.

Page 6 lines 212-213 “and is a valid tool for PCD diagnosis when performed by centers with expertise in its use” it is repeated.

Summary recommendations for PCD diagnosis in North America

Page 7 Paragraph 8 called “Summary recommendations for PCD diagnosis in North America” is quite repetitive and presents points of the ATS recommendations already well described above. The authors could propose a our diagnostic algorithm (Figure 3), which may be helpful for clinical physician.

PCD Foundation CRCN overview

Page 9 lines 309-310 “Over 40 pediatric and adult locations are accredited in the PCDF CRCN (Figure 4).” It would be useful to know the number of accredited centers before and after the publication of the recommendations, to underline the role of the ATS recommendations in improving  diagnostic accuracy. Add another figure with map of PCD Foundation accredited clinical centers before publication of ATS recommendation.
